Classifying software security requirements into confidentiality, integrity, and availability using machine learning approaches

Bagies Taghreed tbagies@kau.edu.sa
Information Technology, Computing and Information Technology, King Abdulaziz University , Jeddah , Saudi Arabia
Alarcon-Aquino Vicente
Electronic publication date: 2024 Nov 26
Publication date: 2024
Volume: 10
Electronic Location ID: e2554
Received 2024 Aug 2; Accepted 2024 Nov 5
Copyright: ©2024 Bagies
Copyright year: 2024
Copyright holder: Bagies
License: This is an open access article distributed under the terms of the Creative Commons Attribution License, which permits unrestricted use, distribution, reproduction and adaptation in any medium and for any purpose provided that it is properly attributed. For attribution, the original author(s), title, publication source (PeerJ Computer Science) and either DOI or URL of the article must be cited.
License URL: https://creativecommons.org/licenses/by/4.0/

Keywords: Security requirements, Software engineering, Sentence-transformer, Machine learning, CIA

Funding: The authors received no funding for this work.

==============================
Security requirements are considered one of the most important non-functional requirements of software. The CIA (confidentiality, integrity, and availability) triad forms the basis for the development of security systems. Each dimension is expressed as having many security requirements that should be designed, implemented, and tested. However, requirements are written in a natural language and may suffer from ambiguity and inconsistency, which makes it harder to distinguish between different security dimensions. Recognizing the security dimensions in a requirements document should facilitate tracing the requirements and ensuring that a dimension has been implemented in a software system. This process should be automated to reduce time and effort for software engineers. In this paper, we propose to classify the security requirements into CIA triads using Term frequency-inverse document frequency and sentence-transformer embedding as two different technologies for feature extraction. For both techniques, we developed five models by using five well-known machine learning algorithms: (1) support vector machine (SVM), (2) K-nearest neighbors (KNN), (3) Random Forest (RF), (4) gradient boosting (GB), and (5) Bernoulli Naive Bayes (BNB). Also, we developed a web interface that facilitates real-time analysis and classifies security requirements into CIA triads. Our results revealed that SVM with the sentence-transformer technique outperformed all classifiers by 87% accuracy in predicting a type of security dimension.

Introduction

One of the essential tasks in the software development life cycle (SDLC) is requirements engineering (RE), which is the process of establishing the system’s services required by a customer (functional requirements) and the constraints under which the system operates and is developed (non-functional requirements). During the RE process, software specification is the process of writing down the user and system requirements in a requirements document. A requirements document is written as natural language sentences. It can be understood by users and customers and may be part of a contract for system development (Sommerville, 2015). The software requirements are essential to ensure the software problem domains are well-defined and address the stakeholders’ needs (Nuseibeh, 2001).

Since functional and non-functional requirements may be mixed in a requirements document, many researchers have proposed several approaches to classifying functional and non-functional requirements within a requirements document (Quba et al., 2021; Dias Canedo & Cordeiro Mendes, 2020; Cleland-Huang et al., 2007; Abad et al., 2017; Navarro-Almanza, Juarez-Ramirez & Licea, 2017; Rahimi, Eassa & Elrefaei, 2020; Slankas & Williams, 2013; Cleland-Huang et al., 2006; Casamayor, Godoy & Campo, 2010). Other researchers have utilized machine learning to extract specific types of non-functional requirements (e.g., security requirements (SRs)) (Kadebu, Thada & Chiurunge, 2018).

The SRs have three essential dimensions called the CIA triad, which stands for confidentiality (CON), integrity (INT), and availability (AVA). According to Sommerville (2015), CON means information in a system may be disclosed or made accessible to people or programs that are not authorized to have access to that information. INT is related to information in a system that may be damaged or corrupted, making it unusual or unreliable. AVA concerns that access to a system or its data that is normally available may not be possible.

The CIA triad forms the basis for the development of security systems. It is used for finding vulnerabilities and methods for creating solutions. It provides a simple yet comprehensive high-level checklist for the evaluation of an effective system satisfying all three aspects. An information security system that is lacking in one of the three aspects of the CIA triad is insufficient. Also, the CIA security triad is valuable in assessing what went wrong—and what worked—after a negative incident. For example, perhaps availability was compromised after a malware attack such as ransomware, but the systems in place were still able to maintain the confidentiality of important information. This data can be used to address weak points and replicate successful policies and implementations (Fortinet, 2024).

Although many researchers have applied machine learning to classify SRs from a requirements document into some categories (e.g., access control, authentications, accountability) (Riaz et al., 2014; Jindal, Malhotra & Jain, 2016; Kobilica, Ayub & Hassine, 2020; Kadebu et al., 2023), none of them have classified SRs into the three triads from a requirements document. In all those works, the dataset was limited (it did not exceed 500 instances and came from one resource). Furthermore, none of them have implemented sentence-transformer embedding. The sentence-transformer embedding technique can capture the semantic textual similarity between sentences and identify patterns in datasets. This might provide better results in classifying SRs (Briggs, 2023).

Our objectives are as follows: (1) to address SRs with their CIA triad at the early stage of SDLC and start with the RE stage to be designed, implemented, and tested, (2) to combine different SR datasets in one set, and (3) to apply sentence-transformer as text-embedding technology to capture the meaning of SRs.

In this paper, we developed models to classify SRs written in requirements documents into CIA. We utilized two classification techniques for feature extraction: (1) term frequency-inverse document frequency (TF-IDF), and (2) sentence-transformer. For comparison purposes, we applied five state-of-the-art machine learning classifiers: support vector machine (SVM), K-nearest neighbors (KNN), random forest (RF), gradient boosting (GB), and Bernoulli naive Bayes (BNB). Also, we built a web interface to provide real-time analysis and classify SRs into CIA triads. We created our dataset by assembling different existing datasets. For dataset preprocessing, we utilized natural language processing (NLP) to clean the data. We evaluated the models by using their accuracy, precision, recall, and F1. The results showed that SVM with the sentence-transformer could predict SRs with an accuracy of 87%.

The main contributions of this paper are:

• Creating SRs dataset gathered from existing SRs datasets.

• Annotating SRs into the CIA triad to be used for different purposes later.

• Comparing two classification techniques for feature extraction: TF-IDF and sentence-transformer.

• Developing five models for both techniques to classify SRs into CIAs.

• Evaluating and analyzing the results of the implemented models.

Related Work

SRs have been addressed in the literature for different purposes and with different proposed techniques. El-Hadary & El-Kassas (2014) proposed a methodology for SRs elicitation for the early integration of security with software development. They constructed a security catalog to help identify SRs with the aid of previous security knowledge. They have made use of evaluation criteria to evaluate the resulting SRs, concentrating on conflict identification among requirements. Jung, Park & Lee (2021) proposed a tool recommending SRs for APT attacks using the Case-Based Problem Domain Ontology specialized for advanced persistent threat attacks. Khanneh & Anu (2022) conducted a literature survey to define SRs engineering and identify the state-of-the-art techniques that can be adopted to impose a prioritization criterion for SRs. Those researchers did not classify or use any machine learning techniques for SRs. In contrast, our work was to develop a model by using sentence-transformer and machine learning techniques to classify SRs into CIA.

Other researchers extracted or identified SRs from different software artifacts or regulatory documents (Riaz et al., 2014; Mohamad et al., 2022). In Riaz et al. (2014), they have developed a tool-assisted process that takes as input a set of natural language artifacts, identifies security-relevant sentences in the artifacts, and classifies them according to the security objectives. They created Riaz’s dataset, which we used as part of our dataset creation. Mohamad et al. (2022) proposed machine learning to identify SRs from regulatory documents. They used TF-IDF and Word2Vec as two techniques for feature extraction. They found that TF-IDF outperformed Word2Vec. They used RF as a machine classifier, and the accuracy was 88.2%. On the contrary, we classified SRs from requirements documents rather than regulatory documents. Instead of Word2Vec, we used a sentence-transformer and compared it with TF-IDF. Our results showed that sentence-transformer provided better results than TF-IDF, and SVM performed better than RF with an accuracy of 87%.

The most related group of researchers to our work have applied machine learning to classify SRs from requirements documents into different purposes and utilized different classifiers (Jindal, Malhotra & Jain, 2016; Kobilica, Ayub & Hassine, 2020; Kadebu et al., 2023). Jindal, Malhotra & Jain (2016) mined the descriptions of SRs in the Software Requirement Specification document and developed four classification models for four types of SRs: (1) authentication-authorization, (2) access control, (3) cryptography-encryption, and (4) data integrity. They used the J48 decision tree method to train the four models and the PROMISE dataset. Each model carried binary classification and corresponded to a specific type of SR. Model#1 corresponded to the ‘authentication-authorization’ type of SRs, model#2 classified ‘access control’ SRs, model#3 classified ‘cryptography-encryption’ SRs, and model#4 classified ‘data integrity’ SRs. The accuracy of model#1, model#2, model#3, and model#4 was 72%, 77%, 69%, and 83%, respectively. Instead of binary classification, we developed a model for multi-class classification and classified SRs into CIA. The highest accuracy for our models was 87%.

Kobilica, Ayub & Hassine (2020) conducted an empirical study to evaluate the performance of 22 supervised machine learning algorithms and two deep learning approaches in extracting SRs from requirements documents. They used the SecReq dataset. Their results show that the long-short-term memory network achieved the best accuracy (84%). On the other hand, our work used the SecReq dataset to increase the dataset size and classify the SRs into CIA by utilizing sentence-transformer. Our results showed that SVM with sentence-transformer achieved higher accuracy (87%) than this study.

Kadebu et al. (2023) proposed a software requirements classification approach considering maintainability as a security requirement. The dataset was collected from students’ project documentation and labeled. It contained 1,317 software requirements, including SRs, functional requirements, and other non-functional requirements. They presented two datasets (one for binary classification of SRs vs. non-SRs and the other for multi-class classification tasks with various more granular SRs vs. non-SRs). They utilized SVM and logistic regression with an average accuracy of 86% in multi-class classification. In addition to SVM, we utilized KNN, RF, GB, and BNB. Our results showed that SVM with sentence-transformer achieved 87%.

To the best of our knowledge, our work was the first to combine all the datasets in the literature for SRs. We utilized the sentence-transformer embedding technique and trained models by using five machine learning classifiers from the literature and adding additional ones. We classified the SRs into CIA triad. The results showed that the sentence-transformer provided better accuracy than the state-of-the-art, which was 87%.

Methodology

Figure 1 provides an overview of our approach. First, we combined various SR datasets from the state-of-the-art and created a comprehensive SR dataset. Then, we preprocessed the dataset by using NLP techniques. The assembled dataset has three cases: (1) data labeled with CIA, (2) data labeled with different security types (e.g., survivability, immunity), and (3) unlabeled data. For the second case, we used the definition of different security types and labeled them with the related CIA. For the third case, we used the active learning approach to annotate the unlabeled data. For feature extraction, we applied two different techniques: TF-IDF and sentence-transformer. Then, we applied five state-of-the-art machine learning classifiers to train models for both feature extraction techniques. For evaluation, we used two approaches: (1) dividing the dataset into 70% for the training data and 30% for the test data, and (2) using 10-fold cross-validation. We utilized accuracy, precision, recall, and F1 to measure the performance of the developed models.

Figure 1 An overview of our approach to developing a model classifying software SRs into CIA.

Security requirements (SRs) dataset gathering

We created the dataset by combining four state-of-the-art SR datasets (Table 1). These datasets were stored in four Excel files (PROMISE_exp, SecReq, DOSSPRE, Riaz). They contained different issues and treated SRs differently. For example, PROMISE_exp labeled SRs with two labels (security and availability). SecReq does not have any label for each SR. DOSSPRE includes ten labels for SRs. Riaz’s dataset has six security objectives that we considered labels.

PROMISE_exp had functional and non-functional requirements and several researchers have used it (Quba et al., 2021; Dias Canedo & Cordeiro Mendes, 2020; Navarro-Almanza, Juarez-Ramirez & Licea, 2017; Kurtanović & Maalej, 2017; Slankas & Williams, 2013; Casamayor, Godoy & Campo, 2010; Cleland-Huang et al., 2007; Abad et al., 2017; Daramola, Sindre & Stalhane, 2012; Jindal, Malhotra & Jain, 2016). It had 86 requirements annotated with Security or Availability (Dias Canedo & Cordeiro Mendes, 2020). We used the availability requirements as they were provided. We labeled the SRs into CON and INT (see “Dataset Annotating”).

SecReq contained three industrial requirements documents: Electronic Purse (ePurse), Customer Premises Network (CPN), and Global Platform Specification (GPS) (Knauss et al., 2021). All listed requirements are related to security. However, they are unlabeled into CIA.

DOSSPRE was a dataset of student software project requirements (Kadebu et al., 2023). It had 515 SRs that were labeled in ten categories (Table 2).

Table 1 Shows the four SRs datasets that we used to assemble our dataset.

Dataset name	Number of SR	Labeling	
PROMISE_exp Mitrevski (2021)	86	Security or availability	
SecReq Knauss et al. (2021)	512	unlabeled	
DOSSPRE Kadebu et al. (2023)	515	10 labels (Table 2)	
Riaz’s Riaz et al. (2014)	22	6 labels (Table 3)	

Table 2 SRs classifications based on Kadebu et al. (2023).

We match each category with a specific CIA triad with respect to its definition.

SR catrgoty	Definition	CIA triads	
Availability (AVA)	System downtime must be kept at a minimum and how authorized users must have access to system resources whenever they need them, notwithstanding disruptions. It is linked to the survivability requirement, which enables maintaining system availability even when it is running in a degraded mode.	AVA	
Authentication (THE)	The system shall be able to verify and confirm correctly the identity of a user wishing to access it. Of importance first is the identification of the user or entity which pertains to creating a unique profile for the user which sets them apart from every other user. This identity must then be confirmed when the user wants to use the system. Only trusted entities are allowed to interact with the system.	CON	
Authorization (THO)	The system shall verify that the user wishing to access a service from the system, has the right to access it. It is also indicated as Access Control. this pertains to issues to do with defining user access rights, role assignment, and privileges. Access is granted to a user according to an Access Control List which can be defined in the system. An entity is authenticated first but that does not define what privileges they have. Authorisation then determines their right in the system.	CON	
Immunity (IMM)	the system shall exhibit an internal capacity to defend itself against attacks and intrusions. It includes enabling input validation, configuration management, and any other means for intrusion prevention. Thus, it is an inbuilt defence mechanism against threats.	INT	
Integrity (INT)	The system is protected from unauthorized modification of data and resources it manages. This includes the prevention of data corruption and tampering with the systemś audit trail. Integrity is guaranteed through hashing, validity checks, and efficient backing up of data.	INT	
Intrusion detection (IND)	The system shall be able to monitor and detect attempts to gain unauthorized access to it. Such attempts should be recorded and alerts generated timeously to prevent damage.	INT	
Confidentiality (CON)	The system shall ensure the privacy of sensitive information and offer personal control over information. This is supported by the authorization requirement, data encryption and data classification capabilities.	CON	
Auditing (AUD)	The system shall monitor and maintain tamperfree records of system activities including system logs, malicious activities, and any system violations. Log files are an important data resource to be well defined and safeguarded.	INT	
Survivability (SUR)	Is evaluated based on the proportion of data objects that remain unaltered and are available for user access.	(Data corruption) INT; others (AVA)	
Maintainability (MAI)	From a security perspective, it specifies the extent to which the system shall prevent any maintenance activities from disrupting any deployed security mechanism.	AVA	

The Riaz dataset contained SRs and sentences from different requirements documents. They classified each SR and sentence into six security objectives (Table 3). Each SR was annotated with more than one security objective. Since all documents, except the Certification Commission for Healthcare Information Technology (CCHIT), in this dataset had sentences extracted from different sections (e.g., introduction), we only used CCHIT containing 331 sentences for SRs. The authors labeled their security impact with (LOW, MODERATE, HIGH). We selected only high and moderate impact and used the first labeled security objective. The number of data collected was 22.

Table 3 SRs Objectives classifications based on Riaz et al. (2014).

We match each category with a specific CIA triad with respect to its definition.

Security objectives	Definition	CIA triads	
Availability	“The degree to which a system or component is operational and accessible when required for use.”	AVA	
Accountability	Degree to which actions affecting software assets “can be traced to the actor responsible for the action”	INT	
Privacy	The degree to which an actor can understand and control how their information is used.	CON	
Integrity	“The degree to which a system or component guards against improper modification or destruction of computer programs or data.”	INT	
Identification Authentication (ACCESS_CONTROL_IDENTITY)	The need to establish that “a claimed identity is valid” for a user, process or device..	CON	
Confidentiality	The degree to which the “data is disclosed only as intended”.	CON	

Dataset preprocessing

The four datasets used were inconsistent as they came from different resources. For example, they had different column numbers and names (serial number vs. ID). Therefore, we created a unified dataset, making it consistent, and cleaned the data.

Unified dataset

To create a unified dataset, we merged all datasets into an Excel sheet. It contained labeled and unlabeled data. For example, rows from PROMISE_exp had labels, whereas rows from SecReq had an empty label. Each dataset had different columns’ names and numbers. For example, some datasets had columns for project ID. We removed this column. Some datasets had a column’s name as a requirement, whereas another dataset had system requirements as a column name. We unified them with the same column name. Some datasets had unnamed columns that had serial numbers. We dropped these columns as they did not have any value to our work.

Data cleaning

To clean the data, we removed special characters (e.g., semicolons, commas, colons, punctuation marks, percentage signs, and asterisks) because they were meaningless and useless. After that, we removed empty rows that did not contain any sentences of SR. As a result, the unified dataset contained 1,135 SRs.

Dataset annotating

At this step, the unified dataset contained 624 labeled instances and 511 unlabeled ones. DOSSPRE had some of its data labeled with CIA. We labeled the other data in DOSSPRE, based on the definition provided by Kadebu et al. (2023) (Table 2). All listed categories were clear and matched the definition, except for the SUR requirements that could be related to INT or AVA. We used the words provided by Riaz et al. (2014) to classify SUR. If, for example, a SUR requirement was related to data corruption, log, or audit files, we annotated it INT. Otherwise, we labeled it AVA.

Similarly, Riza’s dataset included some data labeled with CIA. The other data was labeled with security objectives. We used the provided definition and criteria (e.g., indicative phrases) to annotate them into CIA (Table 3).

The PROMISE_exp had data labeled as security and availability. Similar to what we did on DOSSPRE, we annotated the requirements labeled as security into CON and INT. As a result, the unified dataset contained 624 labeled data and 511 unlabeled data.

Features extracting

Since our dataset was text, we transformed it to be understood by the machine learning algorithms. Our work was the first to utilize sentence-transformer as an embedding technique for SRs. For comparison purposes, we used TF-IDF widely used in the literature.

TF-IDF was aggregated to the sentence level. We utilized the scikit-learn developers (Pedregosa et al., 2011) libraries for text preprocessing. We applied several functions, such as tokenization, stop word removal, stemming, and lemmatization. For the sentence-transformer technique, we used the all-MiniLM-L6-v2” model that uses “siamese and triplet network structures to derive semantically meaningful sentence embeddings that can be compared using cosine-similarity” (Reimers & Gurevych, 2019). The model built features based on text embedding, which helped capture the semantic meaning and context of a text.

Active learning for unlabeled data

At this step, our dataset had 511 unlabeled instances from SeqReq. So, we utilized the active learning approach to automate the labeling process. We used the 624 labeled dataset (from the dataset annotating step) and built an ensemble model containing five classifiers: SVM, KNN, RF, GB, and BNB. We used SVM, KNN, RF, and BNB since they appeared in the literature. Since we had three labels, we should have added at least a classifier to resolve the disagreement between classifiers. Therefore, we included GB (an ensemble technique for building weak decision trees in a way different than RF).

To decide which feature extraction technique, we ran two experiments. In the first experiment, we used TF-IDF and trained five models on 70% of the dataset by utilizing the five classifiers. In the second experiment, we used sentence-transformer embedding and trained five models on 70% of the dataset by utilizing the five classifiers. As a result, sentence-transformer achieved higher accuracy in all classifiers than TF-IDF. Consequently, we used sentence-transformer embedding as feature extraction and trained the five models on all 624 labeled instances. Then, we used them to label the 511 unlabeled SRs into CIA.

We wrote a script to indicate any conflict or disagreement between the five classifiers. If three classifiers agreed with the same label, we accepted this label. If two classifiers or less agreed with the same label, we considered disagreement. An expert annotator resolved this conflict and used the same technique as our data annotating step (see Dataset Annotating).

During the active learning step, there were only five disagreements. As shown in Fig. 2, for example, two classifiers annotated an instance with CON. Two other classifiers labeled the same instance with INT. The last classifier labeled the instance with AVA. Then, the expert labeled these five instances manually.

Figure 2 An example of disagreement between five classifiers during the active learning step.

As a result of this step (annotating the unlabeled instances in our dataset), our dataset had 1,135 labeled instances in CIA. Table 4 summarizes the number of data points associated with its target class. This step was considered an iterative process and enhanced the training model to learn from a broader range of instances. Consequently, it enhanced the capability of models to generalize and improved the performance of the created models.

Table 4 Shows number of data points in our dataset and its target class.

Target class	Number of instances	
AVA	165	
CON	663	
INT	307	

Machine learning classifiers

To classify the SRs into CIA, we developed five classification models (for both TF-IDF and sentence-transformer embedding). We used the same five machine learning algorithms (SVM, KNN, RF, GB, and BNB) that we used during our active learning step. As a result, we built ten models.

Evaluation

We evaluated the developed models by using two techniques (dataset splitting and n-fold cross-validation). For dataset splitting, we shuffled our dataset to make it random such that an instance from PROMISE may be followed by an instance from SecReq. We used a Python function to shuffle the dataset. Then, we divided our dataset into training and test data. We used 70% of the dataset as a training set to develop the models. Then, we used 30% of the dataset to evaluate the models. We used Python libraries and functions to split the dataset (e.g., sklearn.model_selection, train_test_split). Since we fixed the seeding of all models, we did not need to repeat the experiments as they would provide the same results. For n-fold cross-validation, we used 10 folds. We measured the performance of each model based on accuracy, precision, recall, and F1-score. We executed the two experiments by using: (1) sentence-transformer, (2) TF-IDF preprocessing (stemming, lemmatization, stopping word removal), and (3) TF-IDF without preprocessing. In addition, we reported the confusion matrix for all developed models.

Additionally, we used Gradio to develop a web interface facilitating real-time analysis and classification of SRs into CIA triads. The interface allowed users to input SRs. Then, the trained model processed the inputs and predicted their classes. For further analysis, we integrated Llama2, a large language model from Meta AI, providing detailed explanations for each classified SR. It justified why a specific class was assigned to each SR and provided an understanding of the specific requirement’s characteristics.

Results

The results of four performance metrics (accuracy, precision, recall, and F1) are shown in Fig. 3 (experiment#1 splitting dataset into 70% training set and 30% test set) and Fig. 4 (experiment#2 10-fold cross validation). Overall, the sentence-transformer embedding technique enhanced the performance of all classifiers with regard to accuracy, recall, and F1 in both experiments. As shown in Figs. 3A and 4A, the accuracy of all classifiers with the sentence-transformer technique performed better than with TF-IDF. This was because the sentence-transformer captured the semantic meaning of a sentence, while TF-IDF converted the words to numbers and handled them as vector representations. Additionally, SVM had a significant improvement (20% higher) with sentence-transformer than with TF-IDF. The other classifiers had an improvement percentage between 4% and 10% with the sentence-transformer. The BNB classifier had the lowest accuracy for both feature extraction techniques, which was less than 80%.

Figure 3 (A–D) The results of experiment#1: Using 70% training set and 30% test set.

It shows the accuracy, precision, recall, and F1 for the five machine learning classifiers using the TF-IDF and sentence-transformer techniques.

Figure 4 (A–D) The results of experiment#2: Using 10-fold cross validation.

It shows the accuracy, precision, recall, and F1 for the five machine learning classifiers using the TF-IDF and sentence-transformer techniques.

As shown in Figs. 3B, 3C, 3D, 4B, 4C and 4D, SVM with sentence-transformer achieved the highest percentage for all metrics. Although SVM is well-known to work better with binary classification and RF is well-known to work better with multi-classification, using the sentence-transformer enhanced the performance of SVM for multi-classification. The sentence-transformer improved the precision of RF and GB, whereas the precision of KNN and BNB with TF-IDF was better than with the sentence-transformer. The recall and F1 measurements for all classifiers with the sentence-transformer outperformed TF-IDF.

Figures 3 and 4 revealed that applying preprocessing steps to TF-IDF slightly improved the performance of all classifiers in both experiments. Notably, Fig. 3A demonstrated that applying preprocessing steps with TF-IDF on RF significantly improved the accuracy.

For further analysis, Figs. 5, 6 and 7 illustrated the confusion matrices of sentence-transformer, TF-IDF applying preprocessing, and TF-IDF without preprocessing, respectively. The sentence-transformer for all models was better at understanding and predicting the true label than TF-IDF. The TF-IDF techniques could not recognize the CON and AVA target classes, and it classified them as INT. Applying the preprocessing step that included stemming, lemmatization, and stopping word removal with TF-IDF slightly improved the performance of all classifiers.

Figure 5 (A–E) Confusion matrix of all models based on sentence transformer.

Figure 6 (A–E) Confusion matrix of all models based on the TF-IDF with preprocessing.

Figure 7 (A–E) Confusion matrix of all models based on the TF-IDF without preprocessing.

Additionally, the results demonstrated that all models were able to classify INT with a small portion of mistakes. However, all models incorrectly interpreted some AVA as INT. Our dataset contains a small number of instances for AVA such that 15% of data points are AVA. This was not enough to train models compared with other classes (CON and INT).

Furthermore, some SRs were ambiguous and may be interpreted differently. For example, “The system shall allow that if the mobile phone is not present, some authentications are done by the user to obtain access to the system” was provided by DOSSPRE as AVA. However, it should be related to CON. This conveyed some challenges of the nature of the English natural language. In addition, some SRs were ambiguous and may cover more than one CIA. For example, “The system shall be available to secure the files from unauthorised users” could be CON and AVA.

As stated by Sharma (2023), “F1 score is a useful metric for measuring the performance for classification models when you have imbalanced data because it takes into account the type of errors”. As shown in Table 4, our dataset was imbalanced. The F1-score results of all classifiers for 10-fold cross-validation were less than splitting the dataset into 70% and 30% (Figs. 3 and 4D). Consequently, the imbalanced dataset had a negative impact on 10-fold cross-validation over the splitting dataset.

Figure 8 shows the web interface that we created by using LLM. We entered an SR as input. The result displayed the class and the reasoning behind choosing this class.

Figure 8 An example of our created web interface showing a SR as an input and the output.

Discussion

There was a lack of a dataset representing requirements documents from different domains. The existing ones did not reflect all types of software systems. Therefore, we gathered four available datasets from the requirements engineering field.

Since the available datasets had a small number of SRs, we did not apply deep learning classifiers. Deep learning algorithms require a large amount of data.

During the data annotation, one limitation was that we could not find other experts to annotate the 511 unlabeled data points. Therefore, we applied the active learning approach with five well-known classifiers. We trained these models by using 624 instances. Then, we validated the dataset manually with an expert when there was any disagreement between the five classifiers.

The dataset was imbalanced. There were 58% instances for CON, 27% for INT, and 15% for AVA. This could affect the results of our models. Unfortunately, we could not find more SRs that can cover AVA and INT.

To mitigate the imbalance problem, one would suggest using data augmentation to increase the dataset size, which would lead to better accuracy. However, our solution focused more on using sentence-transformer for data embeddings and was compared with TF-IDF as a way of building models classifying SRs into CIA. In addition, all the datasets used did not cover all types of software applications. As part of future work, we would create better datasets that come from different types of software systems for different domains. At this point, it was not beneficial to apply data augmentation for datasets that were not generalized and covered all application domains.

Conclusions and Future Work

We developed five models for TF-IDF and sentence-transformer to classify SRs into CIA from a requirements document. We compared five machine learning classifiers (SVM, KNN, RF, GB, and BNB). SVM with sentence-transformer achieved the highest accuracy, which is 87%. For future work, we will extract SRs from user reviews of mobile applications to build huge datasets that should cover different software system domains and types. Then, we will classify them.

Supplemental Information

Supplemental Information 1 Results from active learning showing the conlfict

Supplemental Information 2 Code for data collection, cleaning, and active learning

Supplemental Information 3 Code for machine learning models, evaluation and LLM for Web interface

Supplemental Information 4 After the dataset annotating step

Supplemental Information 5 Riza’s Dataset

Supplemental Information 6 Raw Dataset (Promise, SeqReq, DOSSPRE) before any annotation or preprocessing

Supplemental Information 7 After the annotated active learning step with an expert resolve the conflict

Additional Information and Declarations

Competing Interests

Author Contributions

Data Availability

The authors declare there are no competing interests.

Taghreed Bagies conceived and designed the experiments, performed the experiments, analyzed the data, performed the computation work, prepared figures and/or tables, authored or reviewed drafts of the article, and approved the final draft.

The following information was supplied regarding data availability:

The dataset and code for annotating and machine learning are available in the Supplemental Files.

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
