# Peer review of "Classifying software security requirements into confidentiality, integrity, and availability using machine learning approaches"

_PeerJ Computer Science, doi:10.7717/peerj-cs.2554_

## Round 0.1 · original submission · Major Revisions

I have received reviews of your manuscript from scholars who are experts on the cited topic. They find the topic interesting; however, several concerns must be addressed regarding experimental results, literature review, methodology, and comparisons with current approaches. These issues require a major revision. Please refer to the reviewers’ comments at the end of this letter; you will see that they advise you to revise your manuscript. If you are prepared to undertake the work required, I would be pleased to reconsider my decision. Please submit a list of changes or a rebuttal against each point that is being raised when you submit your revised manuscript.

Thank you for considering PeerJ Computer Science for the publication of your research.

With kind regards,

Reviewer 1 ·

Basic reporting

no comment

Experimental design

no comment

Validity of the findings

no comment

Additional comments

The objectives of the research are not clearly defined?
The paper not clearly articulate the problem it addresses and the goals of the proposed approach?
The paper is not well-organized and easy to follow.
The methodologies and techniques used in this research are not explained with sufficient details.
The feature extraction methods (TF-IDF, sentence-transformer embedding) and machine learning models are not clearly described?
Proper evaluation must be conducted to show the efficiency of the proposed approach over existing methods?
The accuracy metrics and comparisons between different models are not adequately presented and justified.
The paper must provide a strong rationale for the choice of machine learning algorithms?
The choices of SVM, KNN, RF, GB, and BNB are not justified.
Provide the practical implications of the research.
The paper does not effectively address the challenges of ambiguity and inconsistency in natural language requirements?
The limitations and potential areas for future work are not discussed?

·

Basic reporting

The language in this paper is generally good and easy to understand. The author has provided a clear and professional article structure, with relevant results to the hypothesis. The use of figures, tables, and raw data sharing is also well-done. However, there are a few minor errors, such as the misspelling of "lemmatization" as "limitatization" in Line 182. Overall, the paper is self-contained and provides sufficient field background and context.

Experimental design

The author has performed original primary research within the Aims and Scope of the journal, addressing a well-defined and meaningful research question. The research fills an identified knowledge gap in the classification of security requirements into CIA triads. The investigation is rigorous, performed to a high technical and ethical standard. The methods, including the use of two embeddings and different machine learning classifiers, are described with sufficient detail and information to replicate. The author has also made a good contribution by aggregating four current available datasets into a unified whole, combining different labeling, different columns, and dealing with unlabeled cases.

Validity of the findings

The author has provided robust and statistically sound data, although there are some limitations. The performance of the models using 10-fold cross-validation is generally worse than using a 70-30 train-test split, with a decrease in accuracy, precision, recall, and F1. This is somewhat surprising, as the 10-fold cross-validation would seem to have more training samples (90% of the data in each fold) compared to the 70-30 split (70% of the data for training). I would like to know more about the reason behind this discrepancy.

The dataset also appears to be strongly skewed, with some classes having significantly fewer samples than others. The author has discussed the avoidance of using deep learning models due to the lack of data and the unbalanced dataset, which is a valid concern. I think the author could consider data augmentation or resampling lower represented classes to improve the performance of the models.

Additional comments

Overall, I think the author has done a good job in this paper. I have some minor questions about the use of TF-IDF and sentence embedding for feature extraction, specifically how TF-IDF is aggregated to sentence level or document level, and what level of embedding is used as input to the classifiers. However, these are not major concerns.

Also, the literature review section could be extended to better explain how other works relate to and differ from the author's work, and to highlight the essential novelty that makes the author's work different from existing work. The accuracy of the models is similar to existing works.

---

## Round 0.2 · Minor Revisions

All concerns raised by the reviewers have been addressed satisfactorily; however, the paper still needs further clarification regarding the term "limitatization", cross-validation, and dataset imbalance. These issues require a minor revision. If you are prepared to undertake the work required, I would be pleased to reconsider my decision. Please submit a list of changes or a rebuttal against each point that is being raised when you submit your revised manuscript.

Reviewer 1 ·

Basic reporting

no comment

Experimental design

no comment

Validity of the findings

no comment

Additional comments

I accept this paper.

·

Basic reporting

The author has made improvements, addressing previous concerns about TF-IDF embedding and comparisons with other works. However, the term "limitatization" remains unclear and likely a typo for "lemmatization." This needs clarification or correction. Overall, the paper maintains a clear structure with relevant results and well-organized figures and tables.

Experimental design

The research aligns with the journal's scope and addresses a meaningful question about classifying security requirements into CIA triads. The investigation is thorough, with detailed methods allowing replication. The author's use of sentence-transformer and TF-IDF embeddings, along with dataset aggregation, adds value to the field.

Validity of the findings

While the data is robust, the explanation for why 10-fold cross-validation performs worse than a 70-30 split needs more clarity. The claim that dataset imbalance affects the results suggests addressing this issue might improve performance more than the current method. This calls into question the work's meaningfulness if imbalance is a significant factor.

Additional comments

The author has resolved some earlier concerns, particularly regarding TF-IDF and comparisons with other works. However, the term "limitatization" should be clarified. Also, the author should reconsider the approach to dataset imbalance, as it might offer greater performance improvements and undermine the study's impact.

---

## Round 0.3 · accepted · Accept

I am pleased to inform you that your work has now been accepted for publication in PeerJ Computer Science.

Please be advised that you cannot add or remove authors or references post-acceptance, regardless of the reviewers' request(s).

Thank you for submitting your work to this journal.

With kind regards,